# The Effect of Digital Device Usage on Student Academic Performance: A Case Study

**Maria Limniou** 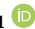

Department of Psychology, Faculty of Health and Life Sciences, University of Liverpool, Liverpool L69 7ZA, UK; Maria.Limniou@liverpool.ac.uk

**Abstract:** The aim of this investigation was to explore student behaviour when students brought their own digital devices into a lecture theatre. A total of 361 undergraduate psychology students from the University of Liverpool who used at least one digital device during lecture time fully completed an online questionnaire (159 first-, 124 second- and 78 third-year psychology students) during the 2018–2019 academic year. Although all the three years of undergraduate students brought laptops and/or smartphones into a lecture theatre, there was no significant difference in academic performance over the years of studies. The findings have linked student multitasking processes in a lecture theatre to Social Cognitive Theory principles (reciprocal interactions between behaviours, learning environment, and individuals). There was a significant difference between the three years regarding the use of applications and student characteristics after controlling for the different types of devices. Students who used only one application during lecture time were more likely to achieve higher academic performance as they were less distracted from their primary tasks of processing and retaining information. Overall, this investigation concluded the importance of reconsidering the teaching delivery process so as to avoid students' escapism using devices during lecture theatres due to their engagement level and lecture norm pressures.

**Keywords:** multitasking; student engagement; student behaviour; distraction; academic performance; Social Cognitive Theory; Bring Your Own Device (BYOD); laptops; smartphones

## 1. Introduction

In recent decades, universities have implemented new infrastructure to allow students to bring their own device (BYOD; bring your own device) for education purposes [1]. Universities have proceeded with the relevant technology infrastructure to support campus-based teaching approaches (e.g., blended and hybrid learning) increasing learning flexibilities such as ease of content access, cost effectiveness, time, and student engagement [2,3]. For example, students are not only allowed to bring their own digital devices into a lecture theatre environment, but they are encouraged to use them in order to support their independent learning from their own home. Teachers perceived the benefits of BYOD to teaching and learning from the perspective of communication, engagement, motivation, collaboration, and potentially tracking students within and beyond the class [4]. Many researchers have studied how students use their devices inside and outside lecture theatres. For instance, students can receive live feedback from their teachers using online (game-based) response systems, increasing student learning engagement during lecture time [5,6]. Students may also use their devices to follow an alternative teaching approach based on flipped classroom [7,8] or they may take notes during lecture time to support their independent learning at home [9]. However, the BYOD practice may be blurring the boundaries of learning time and place [10,11], allowing students to be engaged in multitasking in lecture theatres.

Multitasking processes discuss how humans can carry over more than one independent task at the same time and this is considered as a switching process task. Various

theories have been proposed to discuss multitasking processes, related to the number of tasks that individuals attempt to perform and the limited capacity of the human brain for information processing [12]. For example, multiple resource theory suggests that individuals have a set number of specialised resources that subsume specific functions which are related to cognition and perceptions [13]. Carrier at al. (2009) [14] proposed that people are likely to text or surf the internet whilst listening to music without any decrement in performance, as these two tasks are not competitive (do not share the same resource requirement). The Unified Theory of Multitasking Continuum introduces simultaneous (a potential conflict between tasks) and serial processing (potential inhibition effects on one task as the other is performed) with multitasking processes taking place at different times [15]. Therefore, two or more tasks may compete with one another from the perspective of different resources (similarity with multiple resources theory) and one task will be forced to wait its turn, causing a delay in completion time. Multitasking assumes that individuals somehow have "conscious" awareness and control over the tasks, whereas distractions are usually motivated by sources external to us. Individuals can choose to either ignore the disrupting stimulus or process the distraction. If they choose to do the latter, then potential distractions force individuals to suspend their primary task, leading to a longer time to completion [16].

Many researchers have studied multitasking and distraction processes in a lecture theatre environment and their impact on student performance [17]. For example, they have investigated the impact on the switching process between academic (productive) and non-academic (unproductive) activities on student performance through the use of a laptop during lecture time [18,19]. However, most of the researchers who explored the relationship between multitasking, laptop use in a lecture theatre and academic performance have either included a small number of participants or had a large sample including participants from different disciplines or exposed participants to laboratory and predetermined experimental conditions. Similarly, many other researchers have investigated how mobile devices might be associated with multitasking in classroom/lecture learning environment and student academic performance [20]. It seems that there is a debate regarding the use of mobile phone during lecture time and its association with student academic performance [21,22]. Kuznekoff, Munz and Titsworth (2015) [23] have pointed out that the exchange process of (un)related messages through smartphones during lecture time affects student performance and the note-taking process. The major three distractive ways that mobile phones affect learning performance are related to sources (i.e., notifications, texting process), targets (i.e., messaging has no impact on reading comprehension) and subjects (i.e., information motives, personalities) [24]. On the contrary, Marzouki, Idrissi and Bennani (2017) [25] have mentioned the positive effects of mobile learning on knowledge acquisition, student academic performance, attitudes, and motivation in social constructivist learning environments.

Although researchers have quite extensively explored how, digital device usage in a lecture theatre affects student academic performance [26], only limited attention has been given to learning variables, such as self-efficacy and self-regulation. Self-efficacy refers to individuals' beliefs on their capabilities to perform behaviours that will produce desired outcomes through the accomplishment of specific tasks which then in turn led to achieving one's scholastic goals [27]. Self-regulation consists of social cognitive (e.g., self-efficacy and motivation) and behavioural dimensions (e.g., self-evaluation and effort management) [28,29]. Zhang (2015) [30] has examined the direct and indirect association between learning variables (i.e., self-efficacy, intrinsic, and extrinsic motivation, test anxiety and self-regulation), laptop multitasking and academic performance. The findings of this investigation showed that self-efficacy negatively influenced multitasking, while self-regulation is positively associated with extrinsic motivation (e.g., enjoyment). A recent study has compared the role of self-regulation and self-efficacy in online and face-to-face classrooms on academic performance moderated by gender [31]. The findings of this study argued that females with high self-regulation skills are less exposed to online multitasking experiences and presented higher academic performance compared to males. However,

the participants in this study were enrolled in various courses which might have affected student responses from the perspective of curriculum design along with the usability of courses for their future career.

Limniou, Duret and Hands (2020) [32] have compared learning in regard to student behaviour, learning environment and individual characteristics following Social Cognitive Theory (SCT) to explore multitasking in a lecture theatre and academic performance of first-year students from three different disciplines. The social and cognitive interactions which took place through exposure to the use of digital devices supporting (non-)academic learning activities in a lecture theatre have been studied from the perspectives of student academic performance and their personal characteristics (e.g., self-efficacy, student perceptions of course employability use, test anxiety, and surface strategy). This study argued that there was no difference between academic performance and device usage in a lecture theatre amongst first-year undergraduate students from the same discipline. However, there was a difference between the three disciplines which was mainly related to student characteristics, curriculum design and the teaching delivery process. Applying Social Cognitive Theory (SCT) [33], researchers have explored, in addition to the roles of self-efficacy and self-regulation on academic performance [34], how other individual characteristics (e.g., student background, test anxiety and surface strategy) influenced multitasking processes in a lecture environment. Studying further how the dynamic bond between the interactions of individuals, behaviours and learning environment might be linked to multitasking processes using different digital devices in a lecture theatre, an in-depth understanding may be gained regarding academic performance, learning [35,36] and university BYOD strategies [37,38].

The aim of this study was to identify whether there was a difference in student behaviours when students used either a laptop or a smartphone or both of these devices during lecture time. This study endeavoured to make a connection between student learning and their academic performance to the level of multitasking and distractions in using digital devices in a lecture theatre. To eliminate the risk of including participants with different backgrounds who might have attended different courses (e.g., different curriculum structures) delivered by teachers who did not incorporate the same BYOD policy, this study only focused on students from a specific programme (psychology) across the three years of undergraduate studies. The psychology department has highly encouraged students to bring their own devices into lecture theatres and all teachers have followed the same technological teaching delivery standards. In order to gain a better understanding of student behaviour in relation to the use of digital devices in a lecture theatre, this investigation explored the differences and the associations between undergraduate psychology students from three years of studies in respect to:

- Device usage, multitasking, distractions, and participation in (non-)/learning activities on student academic performance after controlling for the types of devices (laptop, smartphone or both devices); and
- Learning variables, such as self-efficacy, perceived course utility, test anxiety, surface strategy, and behavioural self-regulation, after controlling for the types of devices (laptops, smartphone or both devices).

Figure 1 illustrates how different individual characteristics might link to student behaviour when students used their own digital devices in a lecture learning environment following Social Cognitive Theory (SCT). In this framework, the teaching delivery process in terms of the use of technology in a lecture theatre was not included, as all the university members of staff followed the same technology adoption standards.

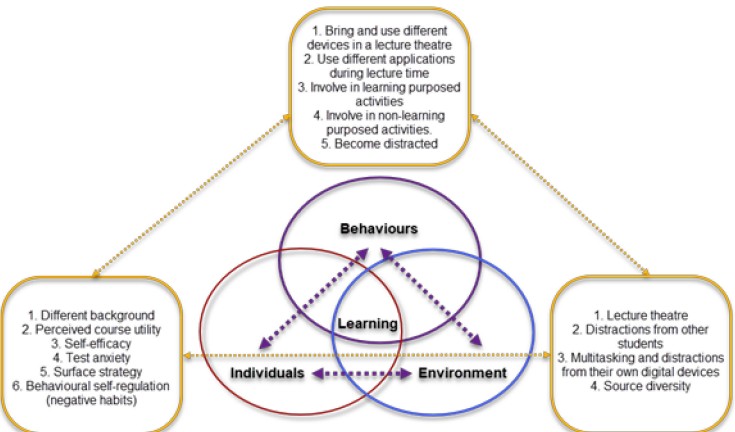

**Figure 1.** Applied Social Cognitive theory (SCT) to a lecture theatre adapted by Limniou, Duret and Hands (2020).

## 2. Materials and Methods

This investigation took place in a psychology department at a research-intensive university in the northwest of England during the 2018–2019 academic year. The psychology curriculum is based on a research–teaching nexus [39], in which within the first two years of studies, psychology students attended the same mandatory modules, whilst in the last year of their studies, they select modules based on their special interest. The psychology department has highly recommended that students bring their own digital devices to lecture theatres, as different digital learning tools (e.g., online voting systems, game-based platforms, and discussion forums) have been integrated into the teaching delivery process. Throughout the programme, a blended learning approach (combining online and/or face-to-face instructional activities inside and outside a lecture environment) has allowed undergraduate psychology students to be actively engaged with online resources and activities before, during or after the lecture time [40]. A variety of assignments (e.g., position paper, essay, oral presentation, and blog preparation) through the semester and final exams (e.g., multiple choice tests, essay type questions and short answer questions) at the end of each semester have supported the assessment process of all the undergraduate modules.

The recruitment process started after this study gained ethical approval from the University Ethics Committee. Initially, a participant information sheet and consent form were provided to participants in order to inform them about the aim of this study, study anonymity and the withdrawal process along with data management details. After their participation in this study, undergraduate psychology students received a debrief sheet.

### 2.1. Questionnare

This was a questionnaire-based study in which the responses of first-, second- and third-year undergraduate psychology students were collected either online or on a paper-based format. The online questionnaire was developed on the web-secured Qualtrics survey platform and was distributed to the first-year students through an internal recruitment scheme, using opportunity sampling as part of the recruitment process. As the second- and third-year undergraduate students do not have access to this internal scheme, a paper-based version of the questionnaire was used, and it was distributed outside their lecture theatres. The recruitment process was based on an opportunistic approach, as students who attended the lecture had the opportunity to participate in this study.

The questionnaire was designed to take approximately 10 min to complete and included 48 multiple-answer items, 7-point Likert scale statements (1 = strongly disagree/not at all and 7 = strongly agree/very great extent), and one open-ended question. This questionnaire is located at the ZENODO repository (https://doi.org/10.5281/zenodo.3333594; accessed on 12 July 2019) and it includes 3 items about device usage, 16 items about learning activities in a lecture theatre, 3 items about distractions from the use of digital devices, 25

items about student learning characteristics (i.e., self-efficacy, surface strategy, behavioural self-regulation) and an open-ended question about the effective use of the digital devices on student learning processes in a lecture theatre. This questionnaire has been previously used to explore the interactions between behaviours, learning environments, and individual characteristics in a lecture theatre for first-year undergraduate students from three different disciplines [32].

### 2.2. Participants

A total of 361 psychology students over the three years of undergraduate studies fully completed the online questionnaire regarding laptop and/or smartphone usage, learning activities in a lecture theatre and their individual characteristics. All the undergraduate psychology students from the three years of studies were recruited on an opportunistic sample basis. Table 1 provides information about participants per year of studies (e.g., gender and average grades). The average age of the sample was 20 ($\pm$1.8). These students were mainly females (81%) and British (94%).

**Table 1.** Student demographics and average grades per year of studies.

| Year of Studies | Number of Participants (Response Rate % per Year) | Males (%) | Females (%) | Grades Mean ($\pm$SD [1]) |
|---|---|---|---|---|
| 1st-Year Student | 159 (47.6% of full cohort) | 20% | 80% | 59.1 ($\pm$8.5) |
| 2nd-Year Student | 124 (29.4% of full cohort) | 16% | 84% | 61.2 ($\pm$6.3) |
| 3rd-Year Student | 78 (20.1% of full cohort) | 22% | 78% | 62.4 ($\pm$7.7) |

[1] SD: Standard Deviation.

## 3. Results

The Statistical Package for Social Sciences (SPSS) software was used for data analysis in this study. Before conducting any statistical analyses, assumptions were tested in order to ensure that there was no violation.

By using ANOVA statistical analysis to compare student academic performance between the three years of studies (Table 1), a significant difference was identified between them ($F_{(2, 496)} = 5.307$, $p = 0.005$, and $\eta^2 = 0.029$). Simple main-effects analysis showed that first-year students had significantly lower grades than third-year student ($p = 0.007$), but there was no significant difference between first- and second-year ($p = 0.07$) and second- and third-year psychology students ($p = 0.5$). A simple regression was carried out to investigate the relationship between student academic performance and year of studies. The regression model was significant, $F_{(4, 359)} = 10,412$, $p < 0.05$, predicted 2.5% of variance ($R^2 = 0.028$). The regression coefficient ($b = 1.66$, 95%, Cl: 0.646 to 2.6) indicted that there is an increase in student academic performance, on average, over the years of 1.66 points.

Table 2 provides an overview of the types of devices that students usually bring to a lecture theatre, the most used applications/software that they utilise to support (non-) learning activities and their behaviours during lecture time. Overall, there was a significant difference between the types of devices that undergraduate psychology students brought into a lecture theatre over the three years of studies, with the majority of the first-year psychology students bringing both types of devices (smartphones and laptop), second-year students bringing mostly laptops, and almost one-half of the third-year cohort bringing only laptops and the other half of the cohort bringing both laptops and smartphones. Regarding the most used applications in a lecture theatre, the second-year students were more actively engaged with social media applications (e.g., Facebook, Twitter) rather than the other two cohorts. Although all cohorts used chat applications to exchange messages with their friends and family members during the lecture, there was a significant difference between student responses, with second-year psychology students more active compared to the other cohorts. Finally, almost half of the first-year students took notes by hand and the rest typed notes on their device, whilst the majority of the second- and third-year students preferred to type on their device. Student responses from the open-

ended questions provided more details about the use of digital devices during lecture time. Students from all the years of studies have mentioned that typing notes on their laptops was an important part of their learning either due to their ability to type notes on devices faster than notes by hand or due to their engagement with the learning process through the typing process. Furthermore, students who "felt bored" with the lecture topic and "felt less engaged" with the delivery process turned their attention to the use of social media, although they were aware of their behaviour's impact on their learning—"When I find a lecture boring, I tend to check social media on my phone which negatively impacts my studies as then I miss important points in the lecture" (second-year student). There were a few students who brought their devices into a lecture theatre just due to "feeling connected to their devices" (third-year student). Digital devices also support students with disability issues (e.g., dyslexia, reading and hearing issues): I use my laptop as I have a hearing impairment. I can record a backup of the lecture in real time whilst writing some basic notes. I then review the recording and make more comprehensive results (first-year student).

**Table 2.** Student responses to questions related to types of devices, used applications and their behaviour in a lecture theatre.

| Behavioural Variable | 1st Year (%) | 2nd Year (%) | 3rd Year (%) | Chi-Square |
|---|---|---|---|---|
| The device(s) that students bring into a lecture theatre | | | | |
| Smartphone | 34.6% | 21.8% | 21.8% | |
| Laptop | 25.8% | 45.2% | 39.7% | $\chi^2(4) = 14.070$, $p = 0.007$ |
| Both devices | 39.6% | 33.1% | 38.5% | |
| The applications that they mostly use during lecture time | | | | |
| Microsoft Word | 58.5% | 59.7% | 57.7% | $\chi^2(2) = 0.084$, $p = 0.959$ |
| Microsoft PowerPoint | 71.7% | 70.2% | 84.6% | $\chi^2(2) = 5.950$, $p = 0.051$ |
| Facebook | 22.4% | 41.9% | 24.4% | $\chi^2(2) = 13.766$, $p = 0.001$ |
| Twitter | 12.6% | 31.5% | 17.9% | $\chi^2(2) = 15.701$, $p = 0.000$ |
| Chat applications | 33.3% | 46.8% | 30.8% | $\chi^2(2) = 7.242$, $p = 0.027$ |
| The behaviour(s) that students usually exhibit during lecture time | | | | |
| Keep notes by hand | 45.9% | 21.0% | 25.6% | $\chi^2(2) = 22.031$, $p = 0.000$ |
| Read PowerPoint slides on their devices | 55.3% | 46.8% | 50.0% | $\chi^2(2) = 2.111$, $p = 0.348$ |
| Type notes on their devices | 52.8% | 68.5% | 61.5% | $\chi^2(2) = 7.263$, $p = 0.026$ |
| Receive and send messages | 40.9% | 54.8% | 39.7% | $\chi^2(2) = 6.733$, $p = 0.034$ |
| Check social media | 39.0% | 49.2% | 28.2% | $\chi^2(2) = 8.939$, $p = 0.011$ |

Table 3 illustrates how the use of different devices (laptops and/or smartphones) and numbers of used applications and behaviours affected student academic performance per year of study. Regarding digital device usage, there was only a significant difference observed between first-year and third-year student academic performance. First-year student academic performance was significantly lower when they used their laptops and both devices (laptops and smartphones) in a lecture theatre compared to third-year student academic performance. However, there were no differences in student academic performance when they used only their smartphones in a lecture theatre. Another interesting point is related to the types of devices and number of applications that were used in a lecture theatre. There is no significant difference between number of applications used by the undergraduate students or the number of different types of behaviours that students exhibited during lecture time regarding their academic performance after controlling for the types of digital devices. However, a multiple regression was conducted to investigate the associations between the different types of devices, the number of applications and the types of behaviours in a lecture learning environment. The regression model was significant and predicted 3.2% of variance (adjusted $R^2 = 0.032$: F (4, 356) = 3.979, $p = 0.004$). In addition to the year of studies which is a positive predictor as presented above, the number of applications mostly used in a lecture theatre was a significant negative predictor

of academic performance (b = −0.133; 95% Cl −1.643 to −0.122; t = −2.282, *p* = 0.023). However, there were no significant predictors of either the types of device (b = −0.004; 95% Cl = 0.928 to 1.078, t = −0.068, *p* = 0.946) or the total of the different behaviours exhibited in a lecture environment (b = 0.043; 95% Cl = −0.499 to 1.104, t = 0.742, *p* = 0.458).

**Table 3.** Student academic performance per year of studies on questions related to student behaviour in a lecture theatre.

| Behavioural Variable | 1st-Year Students (M, SD) | 2nd-Year Students (M, SD) | 3rd-Year Students (M, SD) | ANOVA between the Year of Studies ($\alpha$ = 0.05) |
|---|---|---|---|---|
| Devices mostly used in a lecture theatre | | | | |
| Laptop | 57.7 (±8.73) | 61.4 (±6.58) | 62.8 (±9.16) | $F_{(4, 352)}$ = 1.398, *p* = 0.234, $n^2$ = 0.016 |
| Smartphone | 61.3 (±8.17) | 61.9 (±6.01) | 60.8 (±7.96) | Simple main-effects analysis: a significant difference between 1st and 3rd years (*p* = 0.015) |
| Both Devices | 58.2 (±8.48) | 60.3(±6.15) | 62.8 (±5.63) | No significant difference between 1st and 2nd years (*p* = 0.067) and 2nd and 3rd years (*p* almost equals to 1). |
| Total number of applications which are mostly used during the lecture time | | | | |
| One Application | 60.5 (±8.40) | 60.6 (±6.08) | 63.6 (±7.31) | $F_{(10, 343)}$ = 0.493, *p* = 0.776, $n^2$ = 0.181 |
| Two Applications | 57.9 (±8.57) | 63.1 (±5.35) | 61.5 (±8.19) | Simple main-effects analysis: |
| Three Applications | 58.9 (±7.79) | 60.1 (±6.37) | 61.7 (±9.16) | 1st and 2nd years (*p* almost equals to |
| Four Applications | 56.6 (±8.26) | 60.2 (±6.94) | 64.4 (±4.35) | 1), 1st and 3rd years (*p* = 0.283) and |
| Five Applications | 55.6 (±3.91) | 58.7 (±8.59) | 61.0 (±6.08) | 2nd and 3rd years (*p* = 0.844). |
| Total numbers of different types of behaviours which are mostly exhibited during the lecture time | | | | |
| One behaviour type | 59.6 (±10.81) | 62.9 (±4.51) | 64.4 (±7.21) | $F_{(10, 343)}$ = 1.692, *p* = 0.081, $n^2$ = 0.047 |
| Two behaviour types | 59.7 (±6.65) | 61.8 (±5.48) | 60.3 (±8.71) | Simple main-effects analysis: |
| Three behaviour types | 59.1 (±8.42) | 59.1 (±7.20) | 63.5 (±5.85) | 1st and 2nd years (*p* almost equals to |
| Four behaviour types | 57.8 (±8.64) | 61.4 (±6.50) | 66.2 (±2.59) | 1), 1st and 3rd years (*p* = 0.076) and |
| Five behaviour types | 59.1 (±5.24) | 61.3 (±6.24) | 61.5 (±4.95) | 2nd and 3rd years (*p* = 0.563). |

$\alpha$: the limit of the significant level, M: Mean, SD: Standard Deviation, F(a, b) is the variance value, *p*: significant value, and $n^2$: size effect.

Table 4 illustrates the findings of a two-way ANCOVA statistical analysis between the three cohorts and (non-)learning activities and students' distractions after controlling for the different types of devices. There is no significant difference between student participation in learning activities and academic performance when they brought different types of devices to a lecture theatre. Students who brought only a smartphone to a lecture theatre participated to a lesser extent in learning activities than those students who brought laptops. For example, second-year students who were more engaged with the non-learning activities during lecture time brought mainly smartphones, and became more distracted compared to the other two cohorts.

A multiple regression was conducted to investigate the associations between (non-) learning activities in a lecture theatre along with distractions due to the use of devices over the three years of undergraduate studies. The regression model was significant and predicted 2.7% of variance (adjusted $R^2$ = 0.027: $F_{(4, 355)}$ = 3.031, *p* = 0.011). In addition to the year of studies, which is a positive predictor (b = 0.160; 95% Cl 0.569 to 2.604; t = 3.065, *p* = 0.002), none of the other variables—the number of digital devices (b = −0.019; 95% Cl −1.106 to −0.755; t = −0.371, *p* = 0.710), learning activities (b = −0.025; 95% Cl −0.862 to 0.523; t = −0.482, *p* = 0.630), non-learning activities (b = −0.061; 95% Cl −0.872 to 0.227, t = −1.155, *p* = 0.249) and distractions in a lecture theatre due to the use of digital devices (b = 0.090; 95% Cl = −0.074 to 1.138 t = 1.725, *p* = 0.085)—were significant predictors.

**Table 4.** Comparisons between the student responses per year related to the use of devices for (non-)learning activities and distractions after controlling for the types of digital devices.

| Behavioural and Learning Environment Variable | Year (M, SD) Digital Device (M, SD) | Two-Way ANCOVA between the Years ($\alpha$ = 0.05) |
|---|---|---|
| Learning activities (productive) (13 items, a = 0.883) | 1st Year: 3.9 ($\pm$1.15) Laptop: 4.4 ($\pm$1.03) Smartphone: 3.3 ($\pm$1.18) Both devices: 4.2 ($\pm$0.95) 2nd Year: 3.8 ($\pm$1.31) Laptop: 4.1 ($\pm$1.35) Smartphone: 3.1 ($\pm$1.29) Both devices: 4.0 ($\pm$1.31) 3rd Year: 3.9 ($\pm$1.15) Laptop: 4.2 ($\pm$1.13) Smartphone: 3.3 ($\pm$1.15) Both devices: 4.1 ($\pm$1.02) | $F_{(4, 352)} = 0.581$, $p = 0.677$, $\eta^2 = 0.007$ Simple main-effects analysis: No significant difference between 1st and 2nd year ($p = 0.555$); 1st and 3rd year ($p = 0.339$); and 2nd and 3rd years ($p$ almost equals to 1.000). |
| Non-learning activities (unproductive) (3 items, a = 0.857) | 1st Year: 3.5 ($\pm$1.57) Smartphone: 3.3 ($\pm$1.54) Both devices: 3.6 ($\pm$1.59) 2nd Year: 4.2 ($\pm$1.21) Laptop: 4.0 ($\pm$1.16) Smartphone: 4.4 ($\pm$1.21) Both devices: 4.3 ($\pm$1.26) 3rd Year: 3.3 ($\pm$1.40) Laptop: 3.0 ($\pm$1.31) Smartphone: 3.8 ($\pm$1.71) Both devices: 3.4 ($\pm$1.14) | $F_{(4, 352)} = 0.881$, $p = 0.476$, $\eta^2 = 0.010$ Laptop: 3.4 ($\pm$1.60) Simple main-effects analysis: A significant difference between 1st year and 2nd year ($p < 0.001$); and 2nd year and 3rd year ($p < 0.001$) No difference between 1st and 3rd year ($p$ almost equals to 1). |
| Distractions (3 items, a = 0.685) | 1st Year: 3.1 ($\pm$1.28) Laptop: 3.0 ($\pm$1.13) Smartphone: 3.4 ($\pm$1.37) Both devices: 3.0 ($\pm$1.27) 2nd Year: 3.5 ($\pm$1.33) Laptop: 3.4 ($\pm$1.49) Smartphone: 3.7 ($\pm$1.03) Both devices: 3.5 ($\pm$1.29) 3rd Year: 3.3 ($\pm$1.28) Laptop: 3.2 ($\pm$1.27) Smartphone: 3.2 ($\pm$1.41) Both devices: 3.4 ($\pm$1.26) | $F_{(4, 352)} = 0.471$, $p = 0.757$, $\eta^2 = 0.005$ Simple main-effects analysis:v Significant difference between 1st and 2nd year ($p = 0.045$) No difference between 1st year and 3rd year ($p$ almost equals to 1) 2nd year and 3rd year ($p = 0.562$). |

a = Cronbach's Alpha, $\alpha$: the limit of the significant level, M: Mean, SD: Standard Deviation, F(a, b) is the variance value, $p$: significant value, and $\eta^2$: effect size.

A thematic analysis of student responses to the open-ended question reveals that distractions were mainly related to:

- The smartphone itself: Although students from all the years of studies brought their smartphone into a lecture theatre, they were aware that it caused them more distractions than laptops—"I rarely use my phone during lectures, but I find it very distracting when I get a notification. I don't need to bring my phone to lectures but I feel like I should keep it with me all the time" (first-year student). A second-year student provides more details about the use of smartphones linking it to Fear Of Missing Out (FOMO)—"I know I shouldn't go on my mobile phone when in lectures and that I should be just concentrating on listening and making notes but it's very difficult to disengage from then. I think it's massively due to FOMO, this in turn dramatically affects my ability to multitask and my attitudes on multitasking as I know I can't multitask, but I still try and do. Overall, I fully know that using my phone is distracting but cannot stop the habit". Finally, an explanation as to why

laptops are not so distracting compared to smartphones is provided by a third-year student—"My laptop does not distract me during lectures because it feels too public to being doing private things on like social media etc. but using my phone in the breaks of lectures is a bad idea because it does not actually set me up for the second hour and then it's harder to take notes in the second half because I do not have a proper break, just a phone break". Several students followed techniques to reduce distractions from smartphones such as "I only tend to bring my phone to lectures and I only find this distracting when it is face up, so I place it face down on the desk" (first-year student).

- Their peers' engagement with their devices: Students have mentioned that they were distracted when their peers were involved in non-learning activities due to "constant social media notification tones", "flashing images" from games or "the sound of people typing watching videos i.e., blue planet".

- Learning content and delivery process: Students from all the years of studies pointed out the importance of a more enjoyable way of teaching—"I often cannot always listen for 2 h to heavy content and take it all in and make notes. So I often then turn to my device thinking: I'll catch up on this later as they're talking too fast and won't take it in anyway" (second-year student) and "I struggle to listen to a monotone voice for hours" (third-year student) which prevents them from being "mostly concentrated on the lectures" (first-year student).

Finally, Table 5 illustrates the comparisons between the different individual characteristics in academic performance for the three different years of studies after controlling for the use of different devices in a lecture theatre. There are significant differences in perceived course utility, students' test anxiety, students' surface learning approach and digital learning sources. On the contrary, all the undergraduate students presented a similar self-efficacy and negative habit (behavioural self-regulation) after controlling for the use of different digital devices in a lecture theatre.

**Table 5.** Comparisons between the years of studies, individual learning characteristics and academic performance after controlling for the use of different devices in a lecture theatre.

| Individual Learning Variable | Year (M, SD) | Two-Way ANCOVA between Years ($\alpha = 0.05$) |
|---|---|---|
| Self-efficacy (4 items, a = 0.797) | 1st Year: 4.7($\pm$ 1.08)<br>2nd Year: 4.8 ($\pm$0.77)<br>3rd Year: 4.8 ($\pm$0.95) | $F(2, 357) = 0.408$, $p = 0.665$, $\eta^2 = 0.002$<br>Simple main-effects analysis: no significant difference between all the years ($p$ almost equals to 1). |
| Perceived course utility (3 items, a = 0.748) | 1st Year: 5.8 ($\pm$0.83)<br>2nd Year: 5.4 ($\pm$0.91)<br>3rd Year: 5.5 ($\pm$0.80) | $F(2, 357) = 6.151$, $p = 0.002$, $\eta^2 = 0.033$<br>Simple main-effects analysis: significant differences between 1st year and 2nd year ($p = 0.003$), but no significant difference between 1st and 3rd year ($p = 0.060$) and 2nd year and 3rd year ($p$ almost equals to 1). |
| Test anxiety (4 items, a = 0.855) | 1st Year: 5.5 ($\pm$1.25)<br>2nd Year: 5.9 ($\pm$0.96)<br>3rd Year: 5.8 ($\pm$1.01) | $F(2, 357) = 5.941$, $p = 0.000$, $\eta^2 = 0.032$<br>Simple main-effects analysis: significant difference between 1st year and 2nd year ($p = 0.001$), but no significant differences between 1st year and 3rd year ($p = 0.065$) and 2nd year and 3rd year ($p$ almost equals to 1). |
| Surface strategy (3 items, a = 0.809) | 1st Year: 5.3($\pm$1.22)<br>2nd Year: 5.6 ($\pm$0.95)<br>3rd Year: 5.5 ($\pm$0.98) | $F(2, 357) = 4.056$, $p = 0.003$, $\eta^2 = 0.022$<br>Simple main-effects analysis: significant difference between 1st year and 2nd year ($p = 0.031$), but no significant difference between 1st year and 3rd year ($p = 0.105$) and 2nd year and 3rd year ($p$ almost equals to 1). |
| Source diversity (3 items, a = 0.815) | 1st Year: 5.0 ($\pm$1.04)<br>2nd Year: 5.3 ($\pm$0.93)<br>3rd Year: 5.4 ($\pm$0.94) | $F(2, 357) = 4.681$, $p = 001$, $\eta^2 = 0.026$<br>Simple main-effects analysis: significant difference between 1st year and 3rd year students ($p = 0.017$), but no significant difference between 1st year and 2nd year ($p = 0.113$) and 2nd year and 3rd year ($p$ almost equals to 1). |
| Behavioural self-regulation (negative habit) (7 items, a = 0.837) | 1st Year: 4.7 ($\pm$1.18)<br>2nd Year: 4.8 ($\pm$1.00)<br>3rd Year: 4.6 ($\pm$0.99) | $F(2, 357) = 1.088$, $p = 0.338$, $\eta^2 = 0.006$<br>Simple main-effects analysis: no significant difference between 1st year and 2nd year ($p = 0.642$), 1st year and 3rd year ($p$ almost equals to 1) and 2nd year and 3rd year ($p = 0.812$). |

a = Cronbach's Alpha, $\alpha$: the limit of the significant level, M: Mean, SD: Standard Deviation, F(a, b) is the variance value, $p$: significant value, and $\eta^2$: effect size.

A multiple regression was conducted to investigate the associations between the six variables (perceived course utility, self-efficacy, self-regulation, text anxiety, surface learning and source diversity) with the use of devices over the three years of undergraduate studies. The regression model was significant and predicted 8.5% of variance (adjusted $R^2 = 0.085$: F (8, 352) = 5.164, $p < 0.001$). The year of studies (b = 0.155; 95% Cl 0.494 to 2.574; t = 2.902, $p = 0.004$) and self-efficacy (b = 0.130; 95% Cl 0.090 to 2.017; t = 2.151, $p = 0.032$) are positive predictors, whilst negative habit (self-regulation) (b = −0.190; 95% Cl −2.145 to −0.562; t = 2.151, $p = 0.032$) and digital source diversity (b = −0.131; 95% Cl −2.007 to −0.032; t = −2.031, $p = 0.043$) are negative predictors. Neither of the other two variables, text anxiety (b = 0.052; 95% Cl −0.469 to 1.182; t = −0.850, $p = 0.396$) and surface learning approach (b = −0.129; 95% Cl −0.050 to 1.878, t = −1.864, $p = 0.063$), have an association with student academic performance using different digital devices (b = −0.014; 95% Cl −1.027 to 0.781; t = −0.267, $p = 0.790$) in a lecture environment. The qualitative analysis based on student responses reveals that there are two types of learners: those who used their devices to support their learning process—"Bringing your own device helps to access any content that you need to understand the lecture and makes it easier to go back and reread any points you missed the first time" (first-year student) and those who struggled with self-regulation effort management (negative habit) which was not related to the use of devices—"with or without my phone or device I still find ways to procrastinate" (third-year student).

## 4. Discussion

The aim of this study was to compare undergraduate psychology student behaviours from the perspectives of various learning variables, academic performance, multitasking and distractions, when they used either a laptop or a smartphone or both devices during lecture time. The main assumption of this investigation was related to the course programme. This study assumed that the use of a device might be influenced by curriculum structure and the teaching delivery process. This assumption was based on the findings of a recent study on this area [32] which discussed a comparison between first-year student academic performance in different disciplines. In the current study, participants were undergraduate students from only one discipline (psychology), across all the three levels of their studies, and all students attended 2 hour lectures in a lecture environment. In order to gain a better understanding of the effect of the use of digital devices during lecture time, this study explored the reciprocal interactions of device usage (laptop, smartphone or both devices) with multitasking, distractions and participation in (non-)/learning activities on student academic performance, connecting them to learners' individual characteristics (self-efficacy, perceived course utility, test anxiety, surface strategy, and self-regulation) following Social Cognitive Theory (SCT).

Overall, student academic performance has gradually increased over the years of studies, with first-year psychology students exhibiting lower academic performance compared to the other undergraduate students. On the whole, most undergraduate psychology students used both smartphones and laptops and only a higher percentage of second-year students favoured bringing laptops over smartphones into the lecture theatre compared to the other two cohorts. However, student academic performance did not influence the use of device(s) in lecture theatres and there was an increase in the use of devices in lecture theatres in recent years. During lecture time, undergraduate psychology students used their devices to read PowerPoint lecture slides using their digital device(s) and either took notes by hand or typed notes on their devices to support their independent study time. Students' primary simultaneous tasks were to listen to the lecture, process information presented by teachers and take notes; however, these tasks required mental effort. Farley, Risko and Kingstone (2013) [41] argued that learners' attention to, and retention of, lecture material declined as a function of time on task, increasing the level of mind wandering, suggesting that the use of interventions, such as breaks or a change in lecture pacing, may help. Both quantitative and qualitative analysis of this study showed that students experienced

difficulty in staying focused on the lecture delivery process and they used their device(s) and social media applications to take a break during lecture time and/or to escape from the lecture pacing. Particularly, the second-year students were more active social media users, and they exchanged more messages with their friends/family members during lecture time compared to the other two cohorts. However, no significant difference was found between the years of studies regarding the use of different devices and/or applications and student behaviour during lecture time regarding academic performance. Only a negative association between the number of applications used in a lecture theatre and academic performance was found. This finding might be linked to multitasking theories, as the number of applications might disrupt student attention, forcing them to suspend their primary tasks following when involved in other activities [42]. Primary task completion delays due to disrupting stimulus from the applications predicted lower academic performance. Thus, students who used only one application during lecture time were more likely to achieve a high academic performance, as they experienced fewer disruptions when trying to accomplish their primary tasks of processing and retaining information.

This last finding might also be connected to student involvement in non-learning activities, as a significant difference was found between first- and second-year students and between second- and third-year undergraduate students. The second-year students were highly engaged with social media applications during lecture time, while there was no significant difference between years of studies regarding student involvement in learning activities at any level of studies. Previous studies [30,43,44] have argued that the learning environment influences the frequency and the engagement level of the use of devices. The participants of this study had their device(s) in front of them during a 2 h lecture session and experienced a high frequency of use. Student engagement with their own device(s) was also related to lecture norms and peer influence. For example, the level of the second-year student engagement with non-learning activities was higher than the other two cohorts. Thus, it seems that there was a norm for the second-year cohort supporting student behaviour in checking social media and exchanging messages during lecture time. Social Cognitive Theory (SCT) can also explain the interactions between the learning environment, applications, and peers. The role of SCT regarding the use of different types of devices in a lecture theatre may be supported by main distraction sources based on student responses: 1. the use of smartphones, which distracted their attention; 2. lecture topic and delivery process, which led them to use non-learning applications; 3. their nearby peers' activities. Future work is needed to understand how these reciprocal interactions influence student learning processes under classroom norm pressures.

All participants of this study had a similar background knowledge level as they were from the same discipline (psychology) which required the same entry qualifications and they also had the same opportunities to build up their knowledge over the years by following the same course curriculum. These points explain why the participants exhibited the same level of self-efficacy after controlling for the use of different types of devices (laptops, smartphones, or both devices). Based on the findings of this investigation, the years of studies and students' self-efficacy are positive predictors for academic performance, allowing students to believe that they can perform specific learning behaviours to achieve the desired grade outcome without being influenced by the use of devices in a lecture theatre. However, students' time management skills and procrastination behaviours were negative predictors of student academic performance. This suggests that the more negative self-regulation habits as student had, the lower his/her academic performance. This finding was not related to the use of devices in a lecture theatre, but it was highly related to student learning characteristics. The last point partially contradicts previous findings regarding the self-regulation behaviour dimension which was linked to laptop use in a lecture environment [30]. A potential explanation as to why this study supported a different interpretation might be related to participants. The current study involved participants from the same discipline from all years of studies aiming at connecting their behaviours to their own learning process.

As this study followed SCT principles to investigate the reciprocal interactions between behaviour, learning environment, and individual characteristics, an interesting point emerged from the learning variety of resources which were available to students over the years. Although all students positively evaluated the learning content, a negative association was found with academic performance. Thus, more available learning resources did not support a better academic performance and a significant difference was found between the first- and third-year students after controlling for the different types of devices. This finding might be attributed to student behaviour in typing notes on their device in a lecture theatre. Mueller and Oppenheimer (2014) [45] found that students typed the received information without processing, which prevented them from understanding content during lecture time, while van Wyk and van Ryneveld (2018) [46] discussed the benefits of mobile devices and cognitively demanding note-taking process. Based on the finding of this investigation, this process had an impact on student academic performance as they did not have enough time to process the new lecture information and to process the information from the available variety of resources at the same time. Finally, a significant difference was found between first- and second-year students regarding test anxiety, surface strategy, and perceived course utility, which might be linked to the difference between the cohorts. This was expected, as a difference over the years of studies was identified in first-year students, who showed more social and academic integration and less academic performance [47].

A limitation of this study is that the findings mainly rely on students' self-report responses to a questionnaire. To eliminate the risk of inaccurate findings, qualitative student responses were cross-checked with quantitative responses, but a longitudinal study or use of tracking applications may provide more accurate findings over the years, providing a record of students' actions. Another limitation is related to participant gender, since this study only included undergraduate psychology students from a UK university, who are disproportionately female. More research studies on different disciplines will generalise the findings of this study, allowing educational researchers to explore the difference between gender within the same disciplines. Finally, this study discussed the role of peers and social norms in association with distractions due to the use of digital devices. This might be another limitation as the social norms in a lecture theatre might also be related to the social aspects of learning and the teachers' role. Thus, further work is required to explore how the social perspective of learning can influence multitasking processes.

## 5. Conclusions

Despite the limitations, the aim of this study was to discuss multitasking processes in a lecture theatre as part of the interactions between behaviours, the learning environment, and student characteristics. The findings of this study suggest that although universities and university departments promote the Bring Your Own Device (BYOD) agenda and students are encouraged to use digital devices during lecture time, students are not aware that multitasking, jumping from one task to another, requires high levels of cognitive skills. Student self-efficacy may be overestimated—their behavioural self-regulation skills are highly linked to the use of devices in a lecture theatre and they have not been taught how to self-regulate this. The roots of distractions in a lecture theatre also seem to be related to the learning environment (i.e., peers with smartphones) and learning engagement (i.e., boredom). Thus, teachers should enhance their teaching delivery process through well-supported pedagogical frameworks in order to enhance student learning engagement, while at the same time, they should alert students the consequences of multitasking in their own learning process.

One of the practical implications for this study is that it might avoid misleading conclusions regarding a ban on devices from university lecture environments. Decisions and/or policies should be reconsidered in terms of the teaching delivery process and student engagement, educational challenges in the 21st century. Most universities have the infrastructure allowing students to bring their own devices into a lecture theatre, but the teaching delivery process and student engagement in learning should be reconsidered not

only from a lecture duration perspective (i.e., shorten the 2 hour lecture time and include more breaks) but also in terms of the integration of a variety of resources into the learning and teaching process. Raghuriath, Anker and Nortcliffe (2018) [48] have reached a similar conclusion by exploring the way that teachers integrate technology into a blended learning environment. Their findings led them to recommending that universities further support teachers. The COVID-19 pandemic and necessary changes to the learning delivery process resulted in a departure from traditional methods of teaching, but this may further students' escapism through the use of devices, so it is important to engage students in a learning process that integrates a variety of resources into different teaching modes, supporting active learning processes. One of the theoretical implications of this study is to not only provide information about student learning processes through the use of digital devices, but to provide an example of how a pedagogical theory (Social Cognitive Theory) can be applied to study technology integration into a lecture theatre. Future research on the field of Higher Education pedagogy and technology integration may shed light on the implementation of Bring Your Own Device (BYOD) by universities from the perspectives of student characteristics, learning environment and behaviour through interactions with teachers, resources and peers.

**Funding:** This research received no external funding.

**Institutional Review Board Statement:** This study was conducted according to the guidelines of the Declaration of Helsinki, and approved by the Institute of Life and Human Sciences Research Ethics Committee (School of psychology) of the University of Liverpool (online-based recruitment process protocol code 3497 and date of approval 11/10/2018, and paper-based recruitment process protocol code 4039 and date of approval 15/01/2019).

**Informed Consent Statement:** Informed consent was obtained from all subjects involved in this study.

**Data Availability Statement:** The data that supported the findings of this study are available from the corresponding author, upon reasonable request.

**Acknowledgments:** The author is grateful for the support provided by Destiny Kumari and for her valuable feedback over the revision process.

**Conflicts of Interest:** Author declares no conflict of interest.

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
