# Peer review of "The Effect of Digital Device Usage on Student Academic Performance: A Case Study"

_education, doi:10.3390/educsci11030121_

Round 1

Reviewer 1 Report

The aim of this study was to discuss the multitasking process in a lecture theatre as part of the interactions between behaviours, environment, and students’ characteristics. By comparing psychology undergraduate student behaviours under the perspectives of various learning variables, academic performance, multitasking and distractions, when they used either a laptop or a smartphone or both devices during lecture time. 

The authors have applied a multi-perspective research design that tends to offer interdisciplinary and innovative insights over academic and behavioural performance learning aspects within ICT and in that respect, they are to be congratulated for their time and effort to proceed with such an inter-sectoral design and range of analyses, conceptualisation and discussion performed. 

My only comment would be just to kindly repeat a spell check for minor typos, to reach absolute excellence. 

Please kindly continue such a multi-perspective upper level stream of research!  

Author Response

The author would like to thank Reviewer 1 for the very encouraging comments and suggestions, especially for taking time to review our manuscript despite the pandemic. Following the reviewer’s recommendation, the author has proofread the manuscript thoroughly and has made appropriate amendments. 

Reviewer 2 Report

After analyzing the manuscript entitled "The effect of digital device usage on student academic performance: A Psychology case study", I consider that the study presented is adequate and correct from a general perspective. It presents an adequate theoretical framework and in line with the research, the research method applied is adequate, the results are well presented, the discussion is adequate and the conclusion is adequate. Even so, there are aspects that could be improved, which I list below:

1.- The theoretical framework makes use of an adequate number of references, although the use of references is mostly not from the last 5 years. I therefore recommend including a few more references from the last 5 years. I leave you a manuscript that may be of reference:

https://www.mdpi.com/2072-6643/12/11/3475

2.- The research method is adequate, but I consider that it could be specified a little more, indicating in more detail the process of collecting data and requesting permission to carry out the research.

3.- I recommend the inclusion of a new section, after the conclusions, indicating the theoretical and practical implications generated by this research.

For the rest, I congratulate the authors for their work.

Author Response

The author would like to thank Reviewer 2 for the very constructive comments and suggestions, especially for taking time to review our manuscript despite the pandemic.

The author has found the suggested journal article was very interesting to support the arguments that mobile devices and other digital resources can be used from teachers during the lecture time to enhance student learning engagement. The author has also replaced several other references in order to provide more updated information to the potential readers.

Information about the data collection process and the ethical considerations has been added to the revised version. 

The author has expanded the conclusion section by providing the practical and theoretical implications of this study.

Reviewer 3 Report

The topics addressed in the study are very relevant to the effectiveness of courses delivered in the higher education system. Engagement, concentration of attention are important variables that affect the effectiveness of the educational process. The aim of the study was to highlight the factors that interfere with the learning and teaching process in higher education institutions. The study is also part of an exploration of cognitive processes, such as multitasking. The text has a strong premise. Below it presents some comments that will perhaps contribute to the readability of the study.

  1. First of all start with the title. It is not very clear in the title to use the term psychological. It is too general and unnecessary a statement.
  2. The abstract should include information about the research area (country, institution). In the keywords it is also useful to add missing information about the research area. The year of the research is also welcome in the abstract.
  3. In the introduction, the authors cite old research. For example, Junco's statements from 2012 are quite outdated. Much has changed in the style of new media use over the years. I am thinking here of the speed of the internet, the performance of smartphones, or the style of new media use by young people and young adults.
  4. The research model presented in diagram number 1 is missing an important variable, which is discipline-related determinants. Each lecturer probably has his or her own style of teaching. Thus, there are university staff who prohibit the use of new media in their classes. Why was this aspect omitted from the main research model?
  5. Section 2.2 lacks information on the sampling procedure. What was the selection of students? Purposive? Random?
  6. Did the distribution of data allow the use of parametric tests such as ANOVA, for example?

  7. Before concluding, I propose to make a separate point on the limitations of the methodology used. It is worth using the information from the discussion section here.

    I find the text interesting. It may be useful not only for researchers dealing with the issue of multitasking, but also those conducting research in the field of andragogy or higher education pedagogy. I keep my fingers crossed for corrections.

Author Response

The author would like to thank Reviewer 3 for the very useful comments and suggestions, especially for taking time to review our manuscript despite the pandemic. The author has made appropriate amendments and a reply per comment is provided below.

Comment 1: The author has removed the word “Psychology” from the title. 

Comment 2: The abstract has been revised including information about the institution and the year of study. Additionally, keywords have been added. 

Comment 3: New studies have been added to the revised version. However, several others, mainly related to multitasking and Bandura’s theory, cannot be replaced, as they are considered to be benchmark on this area. Indeed, there are many studies around this topic generally, but many of them explore multitasking process outside the lecture theatre and/or not in Higher Education context. This study has been focused on the learning process which takes place in a University lecture theatre, so the author did not include those which were not highly related to the focus of this topic.   

Comment 4: The Psychology department has highly encouraged all the students and lecturers to use electronic devices in a lecture theatre and the Psychology teachers followed the same technology adoption process, as this is descripted in the Material and Methods section (initial part). The author has highlighted this point in the relevant section. Also, this information has been highlighted to the Material and Methods section in which more details about the curriculum are provided. 

Comment 5: This information was initially provided at the Material and Methods section, but the author has moved them to the Section 2.2. Additional information about the recruitment process was provided in the questionnaire section.

Comment 6: Before conducting any statistical analysis, the assumptions that were related to the statistical analysis were tested. The author has made a relevant comment at the beginning of the Results section. 

Comment 7: The limitation section has been revised. 

Round 2

Reviewer 3 Report

Thank you for the opportunity to review again an interesting study on the implementation of ICT in the learning and teaching process. The text brings a new perspective on the concept of BYOD, which is now very clearly visible and useful from the perspective of the opportunities paradigm of media pedagogy. I recommend the study for publication.